# Protective and vulnerability personality traits associated with PTSD diagnosis after preterm delivery

**Laurane Grand[1]☯, Sabrina Hammami[1]☯, Sarah Bourdon[1]‡, Claudia Demarly Drumelle[1]‡, Julie Auer[1], Anne-Catherine Rolland[2], Julien Eutrope[2]*, Marie Olivier[3]☯**

**1** CHU Reims, Service de Pédopsychiatrie, Reims, France, **2** C2S, CHU Reims, Service de Pédopsychiatrie, Université Reims Champagne Ardenne, Reims, France, **3** C2S, Université Reims Champagne Ardenne, Reims, France

☯ These authors contributed equally to this work.
‡ SB and CDD also contributed equally to this work.
* jeutrope@chu-reims.fr

**Data Availability Statement:** All relevant data are within the manuscript and its Supporting Information files.

## Abstract

Giving birth prematurely is a traumatic event that has many consequences for the mother but also for her baby and their family. Studies have shown that about a quarter of these mothers will suffer from post-traumatic stress disorder (PTSD) as a result. This study aims to identify internal personality factors associated with the development of PTSD in mothers who gave birth before 33 weeks. The results revealed significant correlations between two personality dimensions (neuroticism and extraversion) and the likeliness of developing PTSD in mothers who gave birth prematurely. Neuroticism is positively liked with the disorder while extraversion is negatively correlated with it. Studies should now focus on early detection of PTSD and better interventions for these mothers.

## Introduction

Premature births are becoming increasingly numerous in France and various societal factors can explain this increase: the increase in the average age of bearing the first child, (30.7 years in 2019 compared with 24 years, 50 years ago [1]); the increase in the number of multiple pregnancies linked to the more frequent use of medically assisted reproduction (MAP) [2, 3] and medical and technological advances that have broadened the spectrum of prematurity with higher viability of very premature babies.

Prematurity is defined as a birth before the term of pregnancy, i.e., before 37 weeks of amenorrhea. There are several degrees of prematurity ranging from prematurity (between 33 and 37 weeks of amenorrhea) to extreme prematurity (between 22 and 24 weeks of amenorrhea). According to a 2010 Inserm study [4], it is estimated that approximately 7.4% of births in France are premature. The Epipage [5] study estimates the figure to be between 7% and 12%.

A preterm delivery is often an unexpected event that can be associated with a traumatic experience [6]. This abrupt separation is followed by a stage of symbolic mourning for the

**Funding:** The authors received no specific funding for this work.

**Competing interests:** The authors have declared that no competing interests exist.

imagined baby [7]. Added to this contrast between the imagined baby and the real baby is the difficulty of interacting with a baby who is still immature at all levels. At birth, a full-term baby already has the resources to create a bond with its parents: it can react to their stimuli and interact with them. These reciprocal interactions are rewarding and contribute to the development of the parent-baby bond. The restrictive environment of the premature baby (often in an incubator), as well as its cerebral immaturity, weakens the creation of this bond. To compensate for these constraints, medical teams favor skin-to-skin contact, set up developmental care and train in the psychological support of parents with their newborn. Despite this, at 12 months, 47% of babies born prematurely have insecure attachments compared to 33% of babies born at term [8]. Similarly, at 6 months of age, only 20% of mothers of premature babies show secure attachment representations compared to 53% of mothers of full-term babies [9]. This fragility of the attachment relationship therefore persists over time. Other longer-term consequences have also been noted: distortions in the image of one's child perceived as less autonomous and more sensitive [10], more fragile [11] or even greater anxiety about one's health [12], but also the more frequent appearance of depressive and anxious symptoms in both parents [13].

The link between childbirth and parental stress has been widely studied in research and highlights several phenomena. First, mothers of preterm infants show higher levels of stress and lower estimated quality of life [14] than mothers of full-term infants. Moreover, fathers and mothers seem to differ in the origin of their stress: mothers show stress due to the very fact of their baby's prematurity, whereas fathers show stress depending on the child's health status [15]. In other words, when the child presents high perinatal risks, the parents' stress is similar, but the mother's stress is greater than the father's in the case of a child with low perinatal risks. Finally, studies on parental stress after preterm delivery show that it is associated with the difficulties of the mother-child bond seen above and that it also leads to biological changes. Habersaat et al. [16] showed that babies born preterm whose mothers had high stress had higher cortisol levels than babies born preterm but whose mothers did not have that level of stress. Full-term babies did not appear to be affected by their mothers' anxiety status. Early exposure to high stress would therefore lead to greater vulnerability to environmental stress in infants born preterm. In the mother as well, high levels of stress can lead to long-term biological consequences. Research on trauma has shown that there is a link between traumatic stress and the risk of developing a disease (such as cancer). This is because stress causes premature aging of immune system cells by altering DNA and its repair mechanisms [17]. The DNA strands of people who have experienced a traumatic event are more prone to breakage and their repair processes are more frequent and faster.

High stress can also be a sign of post-traumatic stress disorder (PTSD) in mothers who have given birth prematurely. Although this disorder was once reserved for war trauma, it is now generalized to all types of trauma or stressors. It can also be found in mothers who have experienced a premature birth, in the form of PTSD of the post-partum period. Research on this subject estimates that it occurs in 25% to 33% of mothers after a preterm birth [18, 19]. In this paper, we decided to study the risk factors related to the development of PTSD in mothers who experienced preterm birth. The development of PTSD can have lasting effects on the mother-child bond [11, 20], hence the importance of prevention.

Studies show that the development of PTSD in the mother will exacerbate the difficulties in creating the mother-child bond discussed above [11]. Davies et al. [20] even report on mothers whose PTSD avoidance symptoms caused them to avoid their babies, as the baby embodied the traumatic event. In addition to accentuating the difficulties of creating a mother-child bond, the development of PTSD by the mother will contribute to short- and long-term distortions of her child's image [20]. Indeed, 6 weeks after delivery, mothers suffering from PTSD

describe their baby as "less emotionally warm", "more emotionally demanding" and "more disturbing (invasive)". They also show significantly less desire for closeness with their baby compared to their peers (mothers of full-term babies). The authors hypothesize a defense mechanism of distancing oneself from the child to protect oneself: the child is perceived as a threat and a reminder of the trauma.

All these implications represent an important issue for the detection and early management of these mothers. The prevention of this PTSD, which affects between a third and a quarter of mothers after a premature delivery [18, 19], also raises the question of the vulnerability and protective factors in mothers who develop it.

Research on this subject is numerous and sometimes contradictory. DeMier et al. [21] focused on factors related to the baby (gestational age, weight, Appearance-Pulse-Grimace-Activity-Respiration APGAR score) and obtained positive correlations between severity of prematurity and PTSD symptoms. Conversely, Holditch-Davis et al. [22] found a link between the mother's depressive affects and the development of PTSD but did not find a positive correlation between the severity of prematurity and the development of PTSD.

If studies cannot agree on extrinsic stress factors for the mother (characteristics of the baby) or psychopathological factors (presence of depressive symptoms) to explain vulnerability/protection against PTSD, we can consider intrinsic and more acontextual factors. This is the case for personality and, more specifically, personality traits.

There is currently a consensus to group personality into five dimensions: this is the "Big Five" theory. Costa and McCrae [23] founded a questionnaire using this model, the Neuroticism Extraversion Openness-Personality Inventory Revised (NEO-PI-R), which evaluates five personality dimensions: openness, conscientiousness, extraversion, agreeableness and neuroticism. These are evaluated by a score whose averages, found in this seminal article, show the highest score for agreeableness (which is sensitive to social desirability effects) and the lowest for neuroticism. These personality dimensions are correlated with several psychopathologies. Neuroticism, which is the tendency to experience negative affect, is positively correlated with several psychopathologies such as anxiety disorders or mood disorders [24, 25]. It is also related to vulnerability to stress [26]. Conversely, extraversion, referring to the tendency to experience positive affect, is often associated with a protective personality dimension of stress [26]. The opposite of this theorem also appears to be valid, as a low extraversion score is associated with greater anxiety and greater propensity to suffer from social phobias [25].

In summary, we could imagine that factors intrinsic to mothers influence the propensity to develop PTSD after a premature delivery. However, we know that certain personality traits, notably extraversion and neuroticism, are strongly correlated with the emergence of anxiety disorders. Thus, we hypothesize that these two personality traits may be correlated with the risk of developing PTSD following preterm delivery. More specifically, we expect a significant and positive correlation between neuroticism and the development of PTSD, and a significant and negative correlation between extraversion and the development of PTSD after preterm delivery.

## Materials and methods

### Ethics

This observational study was approved by the Comité de Protection des Personnes Ile de France 8 on October 20, 2021. This approval and a summary of the study were submitted to the French National Agency for the Safety of Medicines. All participants who agreed to take part in this study received an information letter and signed a consent form. This study was conducted in accordance with the good clinical practice as outlined in the Declaration of

Helsinki. Participation in this study was voluntary and patients were free to refuse or to decide at any time not to participate without affecting their relationship with the health care team. Personal data were anonymized. The computerized management of personal data was carried out in accordance with the reference methodology of the French National Commission for Information Technology and Civil Liberties.

## Participants

One hundred and forty-four preterm births (<33 weeks of amenorrhea) took place at the Reims University Hospital between January, 2018 and April, 2019. Fifty-one participants met the inclusion criteria and agreed to participate in this observational study. Each participant was given a document explaining the aim of the study as well as the contact details of the medical supervisor in charge of this study. They were also asked to sign two identical consent forms and keep one of them. All participants delivered a preterm infant (<33 weeks of amenorrhea) without associated pathology. Mothers whose babies died were not included in the study (15 deaths). All participants were over the age of 18 years, were native French speakers, and had no known psychiatric pathology.

## Procedures

The participants were met at the University Hospital of Reims in the neonatal intensive care unit, in the 5th week following their delivery. The research study was explained to them and an appointment was made with their consent. Participants were informed that all data would remain anonymous.

## Material

The collection of socio-demographic data was proposed to the mothers and allowed us to establish initial contact while recording certain elements of the trauma. Once these data had been collected, the mothers answered the PPQ questionnaire. Finally, we asked them to answer the 60 questions in the Neuroticism Extraversion Openness Five Factor Inventory (NEO-FFI).

## Collection of socio-demographic and clinical data

Various socio-demographic and clinical data concerning the mother and the preterm infant were collected in order to evaluate the influence of these factors on our model of vulnerability/ protection to post-traumatic stress after preterm delivery. Thus, the mother's age, marital status, level of education, and occupation were reported. For the mothers, the clinical data concerned: gestational age, parity, obstetrical history, psychiatric history, pregnancy history (preterm delivery [PAD], hospitalization, mode of delivery, multiple pregnancy, anesthesia, perceived support from the spouse and family, post-delivery debriefing with a midwife). Clinical data for the baby included term, weight, and presence or absence of intrauterine growth restriction (IUGR).

## The Perinatal PTSD Questionnaire

The Perinatal PTSD Questionnaire (PPQ) is a self-administered questionnaire for the diagnosis of post-traumatic stress disorder in the specific case of preterm birth. It was created by DeMier et al. in the United States and translated into French in 2004 by Pierrehumbert et al. [19]. It consists of 14 statements that the patient confirms or refutes. If the patient answers "yes" to six or more statements, she is considered to be suffering from PTSD.

### The Neuroticism Extraversion Openness Five Factor Inventory (NEO-FFI)

The NEO-FFI is the short version of the NEO-PI-R. Sixty questions allow an individual to be placed on five personality dimensions of Costa and McCrae's five-factor model (openness, conscientiousness, extraversion, agreeableness and neuroticism). This self-questionnaire, although it does not assess all facets of each dimension as does the NEO-PI-R, has the advantage of being shorter and therefore requires less time and cognitive cost on the part of participants. In order to answer, participants are asked to rate themselves on a 5-point Likert scale ranging from "strongly disagree" to "strongly agree".

### Statistical analysis

Statistical analyses were performed using Statistica software (TIBCO Software Inc., Palo Alto, California, United States of America). To test our hypotheses that there is a link between the personality dimension and the propensity to develop PTSD, Pearson's correlations were performed, given that the application conditions were met. To compare the means of the categorical data between our two groups (groups with PTSD *vs.* group without PTSD), Student's t-tests were performed. Finally, to test the weight of personality dimensions in predicting the development of PTSD, multiple regression was performed with PPQ score as the dependent variable and personality dimensions as the independent explanatory variables. The significance level was set at $p < 0.05$.

## Results

Before testing our hypotheses, we looked at the demographics of the mothers included in our study who were divided into two groups determined by their PPQ score with a cut-off score greater than or equal to 6: mothers with PTSD *vs.* mothers without PTSD (Table 1). These statistics allowed a first look at the data before starting the analyses. Of note, there were 16 mothers out of 51 (31.37%) of mothers with PTSD more than 1 month after their delivery, which is in line with data in the literature [18, 19] (Table 1). The two groups did not differ significantly in any of these demographic data.

The descriptive statistics of the personality dimensions are also consistent with data observed in the literature [23] (Table 2).

As to test the internal consistency of the different dimensions, we have proceeded to the calculations of Cronbach's alphas. As shown in Table 3 they went from 0.67 to 0.86 for the big five dimensions and the PPQ. Thus, internal consistency of our questionnaires is very good.

**Table 1. Demographic data.**

|  | Group of mothers with a PTSD | Group of mothers without a PTSD |
|---|---|---|
| (%) | 16 (31.7%) | 35 (68.63%) |
| Mean age of mothers (sd) | 28.7 (6.54) | 30.25 (5.95) |
| Mean gestational age of babies (sd) | 28.46 (2.29) | 28.61 (2.34) |
| Working mothers (%) | 9/16 (56,25%) | 21/35 (60%) |
| Mean level of education (sd) | 12.06 (2.62) | 13.43 (2.44) |
| Mothers in a relationship (%) | 16/16 (100%) | 32/35 (91.43%) |
| First-time mothers (%) | 11/16 (68,75%) | 25/35 (71.43%) |
| Multiple pregnancy (%) | 4/11 (36.36%) | 11/35 (31.43%) |

(sd): standard deviation

**Table 2. Descriptive statistics of personality dimensions and PTSD.**

|  |  | Mean | Standard deviation (sd) | Min | Max |
|---|---|---|---|---|---|
| PPQ | PTSD Group | 7.19 | 1.33 | 6 | 10 |
|  | Control Group | 2.69 | 1.75 | 0 | 5 |
| NEO-FFI | Neuroticism | 20.67 | 9.37 | 1 | 39 |
|  | Extraversion | 29.55 | 6.23 | 9 | 44 |
|  | Openness | 22.9 | 6.75 | 10 | 40 |
|  | Agreeableness | 33.04 | 6.19 | 17 | 47 |
|  | Conscientiousness | 36.39 | 6.47 | 19 | 47 |

To test the hypothesis that there is a positive correlation between neuroticism and the propensity to develop PTSD, we considered the neuroticism score on the NEO-FFI and the PPQ score. The results indicate a positive correlation between the neuroticism score and the PPQ score ($r = 0.38$, $p < 0.01$), (Fig 1). Our first hypothesis of a positive link between neuroticism and the propensity to develop PTSD was therefore validated.

We then tested our second hypothesis of a correlation between extraversion and propensity to develop PTSD. We found a negative correlation between the extraversion score and the PPQ score ($r = -0.31$, $p = 0.025$), (Fig 2).

For exploratory purposes, we also sought to investigate the relationships between PPQ scores and other personality dimensions of the NEO-FFI, as well as between PPQ scores and socio-demographic data.

When correlations were tested on openness, conscientiousness and agreeableness, no significant results were obtained. These personality dimensions are therefore not likely be related to the probability of developing PTSD after preterm delivery.

We then performed correlation analyses and comparisons of means by considering the following numerical socio-demographic data with the PPQ: baby's birth weight, baby's gestational age in weeks of amenorrhea, gestational age, parity, number of babies, education level, and mother's age. We also addressed dichotomous categorical data such as occupation, presence or absence of IUGR, satisfaction with spousal and family support, experience of threatened PAD, and hospitalization. Of these correlation analyses and comparisons of means, some indicated a trend but none were significant.

We then compared the two groups (mothers with PTSD *vs.* mothers without PTSD) by considering binary dichotomous data such as history of depression or previous drug treatment. We obtained a significant difference ($p = 0.022$) when considering the history of depression with a significantly higher mean PPQ score for mothers who had suffered from depression in the past (Table 4).

In other words, having a history of depression would make one more vulnerable to PTSD in the face of a traumatic event such as preterm delivery. To ensure that this is not simply a

**Table 3. Cronbach's alphas.**

| Questionnaires | | | Cronbach's alpha |
|---|---|---|---|
| **PPQ** | | | **0.67** |
| NEO-FFI | | Neuroticism | 0.86 |
|  | | Extraversion | 0.75 |
|  | | Openness | 0.70 |
|  | | Agreeableness | 0.69 |
|  | | Conscientiousness | 0.83 |

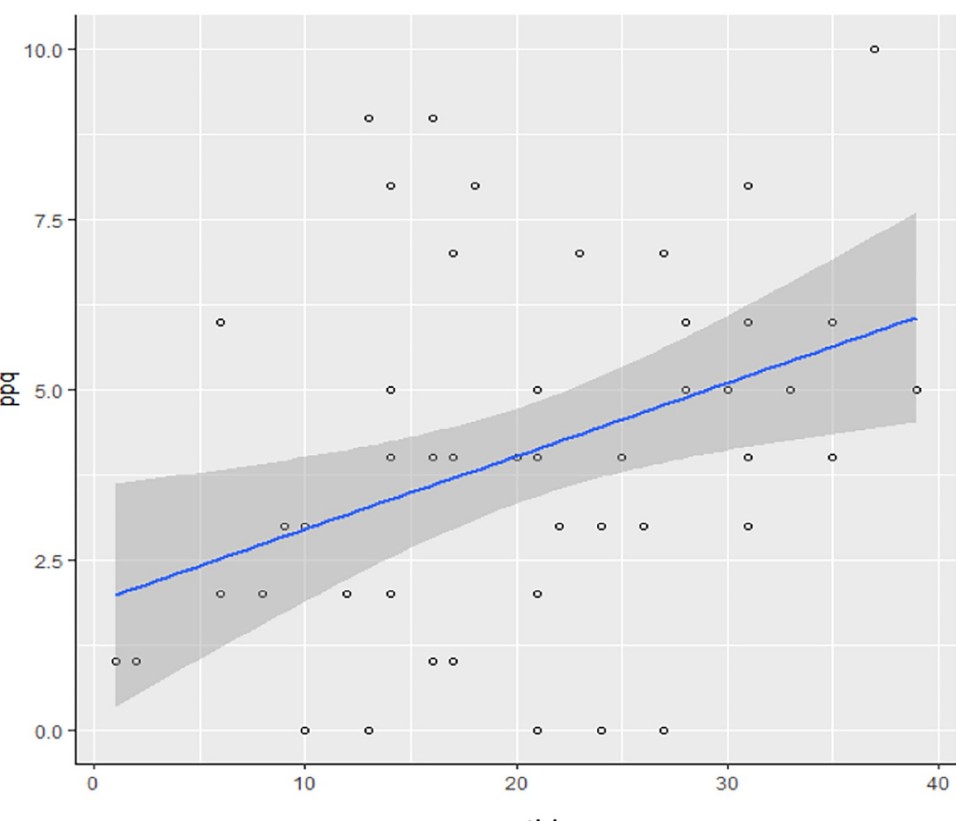

**Fig 1. Correlation between neuroticism and PPQ.**

result explained by the Neuroticism score, we performed a Student's t-test to compare the two groups by neuroticism score. No significant difference was found. The history of depression therefore explains the vulnerability to PTSD independently of the Neuroticism score.

Finally, our model indicates that 23% of the variance of the PPQ score is explained by personality dimensions. Multiple regression analyses highlight the explanatory variables, neuroticism and extraversion, with significant betas of $b^* = -0.31$ and $b^* = 0.34$, respectively, but an $R^2$ of 0.23 in particular (Table 5). Among the five personality dimensions, these variables are therefore the only predictors of PTSD. Our correlation analyses therefore revealed a link between neuroticism and extraversion and the propensity to develop PTSD in mothers who gave birth prematurely before 33 weeks.

Our two hypotheses are therefore validated. Multiple regression analyses also indicated that the personality dimensions predict almost a quarter of the probability of developing PTSD.

**Table 4. Comparison of means between PPQs and history of depression.**

| | Mothers with a history of depression | | Mothers without a history of depression | | p |
|---|---|---|---|---|---|
| | mean | Standard deviation | mean | Standard deviation | |
| PPQ | 5.8 | 1.75 | 3.68 | 2.68 | 0.022 |

Cohen d value: .83 with confidence intervals 95% [0,32; 3,92]

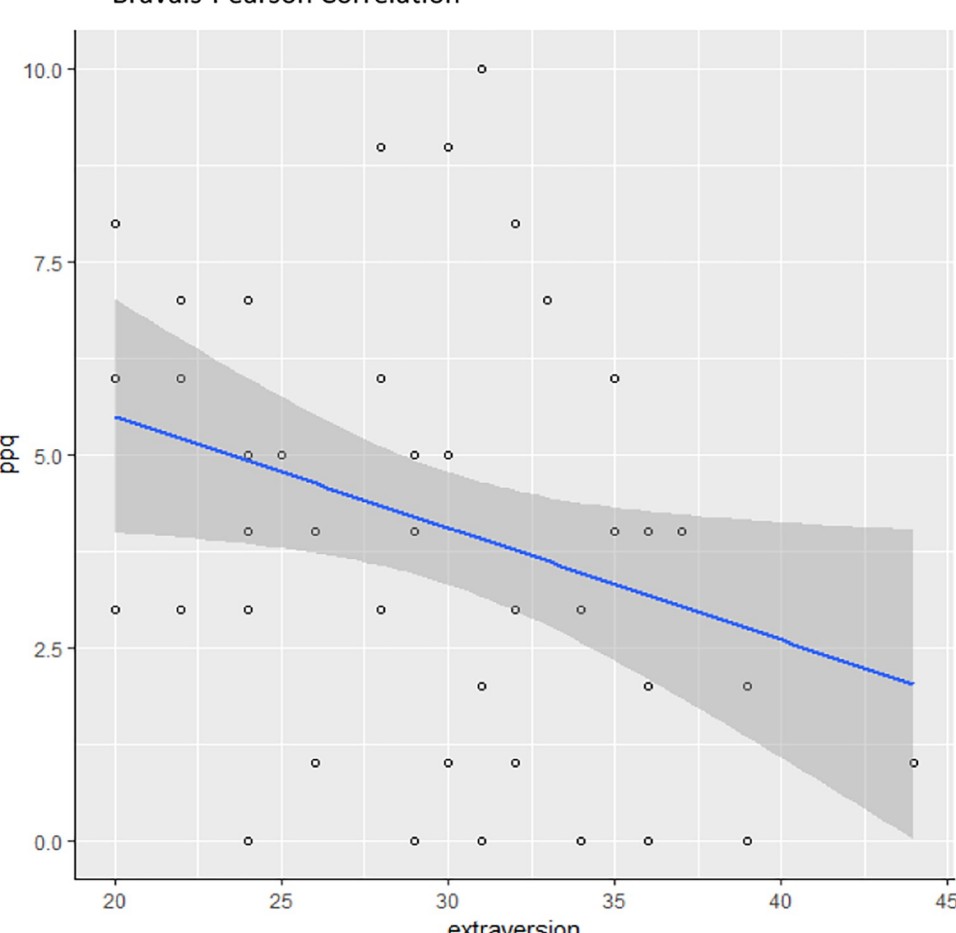

**Fig 2. Correlation between extraversion and PPQ.**

**Table 5. Multiple regression.**

| Predictors | Betas (b) | Unstandardized b | p | $R^2 = 0.23$ |
|---|---|---|---|---|
| Neuroticism | 0.34* | .10* | < 0.02 | |
| Extraversion | -0.31* | -.13* | < 0.05 | |
| Conscientiousness | 0.18 | .07 | | |
| Openness | 0.13 | .05 | | |
| Agreeableness | -0.1 | -.00 | | |

*The beta is significant

## Discussion

Two links between personality and the likelihood of developing PTSD emerge from this study: neuroticism is a vulnerability factor for PTSD and extraversion is a protective factor for PTSD. These results are consistent and in line with the current literature on personality dimensions.

Neuroticism is consistently correlated with several psychopathologies such as anxiety disorders and mood disorders [24]. A high neuroticism score may reflect psychological

vulnerability (conversely, a low score indicates emotional stability). Vulnerability to stress, which is a facet of this personality dimension, may explain some of the correlation with PTSD. If we wanted to isolate and measure the impact of each facet of neuroticism (anxiety, anger/hostility, depression, social shyness, impulsivity, and vulnerability to stress) we would have to use its long version, the NEO-PI-R.

Extraversion, which refers to the propensity to feel positive effects, is found in individuals who are more open, seek out the company of others more, and enjoy lively environments. It includes the following six facets: warmth, gregariousness, assertiveness, activity, sensation seeking, and positive emotions. These individuals are also more likely to ask for help and to externalize their experiences through speech. Research has already proven the beneficial effects of speech and the ability to elaborate a traumatic event to transform the implicit imprint of the trauma into a coherent narrative in PTSD patients [17, 27]. The tendency to feel positive emotions as well as the tendency to reach out to others arguably explains, in part, our negative correlation with PTSD. Mothers with PTSD are therefore more frequently mothers who will not spontaneously reach out to others. We were surprised to find that eight mothers out of 51 reported having had an explanatory debriefing on the conditions of their delivery with a midwife. It is therefore possible that a systematic debriefing of mothers who have had a preterm birth with a health professional could make it possible to detect those who are likely to develop PTSD in the following weeks, for prevention purposes.

Although these results are consistent with our expectations based on the literature, this research highlights the importance of personality dimensions in the probability of developing PTSD. We tested all five personality traits in our study but as expected, only Extraversion and Neuroticism showed a correlation with the risk of PTSD diagnosis in mothers after preterm birth. Indeed, in addition to the correlation coefficients obtained, multiple regression analyses indicated that the development of PTSD after preterm delivery is 23% explained by two personality dimensions.

Furthermore, we found a positive correlation between the probability of developing PTSD after preterm delivery (<33 weeks) and the presence of a history of depression. This could be explored in further research as it would be interesting to better understand this correlation. Several hypothesis could be investigated: depression and anxiety are underpinned by similar cerebral mechanisms and thus one who is vulnerable to depression would also be more vulnerable to anxiety and PSTD; Experiencing an episode of depression would make one more vulnerable to mental pathologies; vulnerability factors can explain both the vulnerability to depression and to PTSD. According to our paper, we could also argue that having a high score in Neuroticism makes one more vulnerable to both depression and anxiety pathologies.

## Limitations

Certain limitations to this research must be acknowledged. First, the sample of this observational study is not large enough to replicate the results highlighted by other research, such as the impact of the baby's health status at birth on maternal trauma [28]. We also decided not to include mothers whose child died for practical, ethical and clinical reasons. Women whose baby passed are not likely to stay at the hospital and will often go home shorty after delivery which makes it harder for us to include them. The clinical symptoms women who lost a child experience also differ from women whose child survived preterm birth and could be addressed in a further research.

Furthermore, we used self-reported measures to assess personality traits (as they are the most universally used) but they are subject to bias, especially social-desirability bias. We also could have used the NEO-PI-R test rather than the NEO-FFI, in order to isolate the impact of

the facets of the personality dimensions. Finally, as personality dimensions explain 23% of the variance, other explanatory factors, such as history of depression and current depressive symptomatology, should be considered. As an example, the stress of mothers was assessed with the PPQ after delivery but it could have been measured before birth as well to evaluate the impact of preterm birth on it. It was not done in our study for obvious methodological reasons as we could not meet every pregnant woman, not knowing if they will give birth prematurely or not.

### Clinical and research perspectives

These results open up many avenues for clinical research.

In keeping with the idea of the effect of other explanatory variables, it might be interesting in a future study to introduce a depression scale. Some studies, like ours, have found a link between depression and PTSD [22] and others between depression and personality traits [24, 25], which we did not find here. This variable could thus be a mediating variable between personality and PTSD.

In addition, it is necessary to study the clinical applications that could be beneficial to patients, particularly by knowing the harmful consequences of maternal PTSD on the mother-child bond [20]. Thus, we could imagine systematizing prevention methods in maternity hospitals by proposing that pregnant women complete the NEO-FFI questionnaire and a depression questionnaire such as the BDI during their pregnancy follow-up. A history of depression could be systematically collected during the medical follow-up of the pregnancy. Mothers at greater risk could thus benefit from more specific follow-up. Moreover, because of the negative correlation between the extraversion score and the propensity to develop PTSD, it would be interesting to encourage each mother who has given birth prematurely to express herself regularly on the particular conditions of her delivery in the weeks following this potentially traumatic event. We could also imagine replicating this research with a group of mothers whose baby was not a preterm baby in order to better identify and prevent risk of PSTD for every woman.

Finally, Eye Movement Desensitization and Reprocessing (EMDR) therapies [29], which have proved their worth in trauma management, could also be considered after preterm delivery. In 2010, Lapp et al. [30] compared various types of treatment after a traumatic (non-preterm) delivery and obtained convincing results for Cognitive Behavioral Therapy (CBT) and EMDR treatment. The value of these therapies could be evaluated in the context of preterm delivery. It would thus be possible to consider that a few sessions of EMDR therapy in the immediate aftermath of preterm delivery could have a significant effect on the incidence rate of PTSD in the month following delivery.

## Supporting information

**S1 Data.**
(PDF)

## Acknowledgments

We would like to thank everyone who made this research possible from the hospital of Reims who gave its authorization to all the patients who accepted to be part of it. We would also like to thank the editors of *PLoS ONE* and the reviewers for their thoughtful comments that helped us improve this article.

## Author Contributions

**Conceptualization:** Laurane Grand, Marie Olivier.

**Investigation:** Sabrina Hammami, Sarah Bourdon, Claudia Demarly Drumelle, Julie Auer.

**Methodology:** Laurane Grand, Sabrina Hammami, Marie Olivier.

**Project administration:** Anne-Catherine Rolland, Marie Olivier.

**Supervision:** Marie Olivier.

**Validation:** Julien Eutrope, Marie Olivier.

**Writing – original draft:** Laurane Grand.

**Writing – review & editing:** Julien Eutrope.

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
