## [Decision Letter · Decision Letter 0]

2 Feb 2023

PONE-D-22-23913Protective and vulnerability factors for PTSD after preterm deliveryPLOS ONE

Dear Dr. Eutrope,

Thank you for submitting your manuscript to PLOS ONE. After careful consideration, we feel that it has merit but does not fully meet PLOS ONE’s publication criteria as it currently stands. Therefore, we invite you to submit a revised version of the manuscript that addresses the points raised during the review process.

We look forward to receiving your revised manuscript.

Kind regards,

Charles Martin-Krumm, Ph.D.

Academic Editor

PLOS ONE

and https://journals.plos.org/plosone/s/file?id=ba62/PLOSOne_formatting_sample_title_authors_affiliations.pdf.

a) Did participants provide their written or verbal informed consent to participate in this study?

Reviewers' comments:

Reviewer's Responses to Questions

**Comments to the Author**

1. Is the manuscript technically sound, and do the data support the conclusions?

Reviewer #1: Partly

2. Has the statistical analysis been performed appropriately and rigorously? 

Reviewer #1: I Don't Know

3. Have the authors made all data underlying the findings in their manuscript fully available?

Reviewer #1: No

4. Is the manuscript presented in an intelligible fashion and written in standard English?

Reviewer #1: Yes

5. Review Comments to the Author

Reviewer #1: Thank you very much for the work you have done.

I don't have any particular recommendation for your introduction your discussion and your conclusion which in my opinion was well done.

On the other hand, your result part has for me several shortcomings.

First, the Cronbach's alphas of your scales in your study are missing. This is a problem since we cannot guess the minimum psychometric properties of your scales in your sample.

Second, you present some results in your tables and you do not present the data for your multiple regression apart from the explained variance. This forces the reader to take your word for it.

I am asking for a major revision so that you can rewrite this section and present the results according to research standards. Otherwise, your paper will not be published.

I know that it is frustrating and complicated to revise part of your work, but your work is important to the field, and without these elements, it loses its strength.

Good luck, I'm sure you'll get through it.

6. PLOS authors have the option to publish the peer review history of their article (what does this mean?). If published, this will include your full peer review and any attached files.

Reviewer #1: No

---

## [Author Response · Author response to Decision Letter 0]

24 Mar 2023

Dear Reviewer,

Thank you for your recommendation on our paper. We really appreciate the time and consideration you put into our work. Here are our answers to the questions and remarks you raised:

1. Please ensure that your manuscript meets PLOS ONE's style requirements, including those for file naming

We have edited our document to meet the PLOS ONE’s style requirements in terms of font and titles.

2. Please amend your current ethics statement to address the consent form:

We have added a few information addressing this question in our paper in further details. Yes, participants received a written document that they signed. This question was addressed in the ethical part, we added it in the participant paragraph as well following your concerns.

3. In your Data Availability statement, you have not specified where the minimal data set underlying the results described in your manuscript can be found. PLOS defines a study's minimal data set as the underlying data used to reach the conclusions drawn in the manuscript and any additional data required to replicate the reported study findings in their entirety. All PLOS journals require that the minimal data set be made fully available.

We see no restriction in making our data public. It was a mistake on our end not to share them. All relevant data are within the manuscript and its Supporting Information files.

Comments to the author:

We added tables and all our data to strengthen our paper as per recommended. We added the multiple regression table, the Cronbach’s alphas of our scales and all data used to lead to our results. 

Again, thank you very much for your feedback and recommendation. We hope we were able to improve our paper to meet your Journal’s requirements and contribute to new findings in our field. 

Best regards,

---

## [Decision Letter · Decision Letter 1]

13 Nov 2023

PONE-D-22-23913R1Protective and vulnerability factors for PTSD after preterm deliveryPLOS ONE

Dear Dr. Eutrope,

Thank you for submitting your manuscript to PLOS ONE. After careful consideration, we feel that it has merit but does not fully meet PLOS ONE’s publication criteria as it currently stands. Therefore, we invite you to submit a revised version of the manuscript that addresses the points raised during the review process.

We look forward to receiving your revised manuscript.

Kind regards,

Giuseppe Marano

Academic Editor

PLOS ONE

Reviewers' comments:

Reviewer's Responses to Questions

**Comments to the Author**

1. If the authors have adequately addressed your comments raised in a previous round of review and you feel that this manuscript is now acceptable for publication, you may indicate that here to bypass the “Comments to the Author” section, enter your conflict of interest statement in the “Confidential to Editor” section, and submit your "Accept" recommendation.

Reviewer #2: All comments have been addressed

Reviewer #3: All comments have been addressed

Reviewer #4: (No Response)

2. Is the manuscript technically sound, and do the data support the conclusions?

Reviewer #2: No

Reviewer #3: Yes

Reviewer #4: Yes

3. Has the statistical analysis been performed appropriately and rigorously? 

Reviewer #2: No

Reviewer #3: Yes

Reviewer #4: Yes

4. Have the authors made all data underlying the findings in their manuscript fully available?

Reviewer #2: (No Response)

Reviewer #3: Yes

Reviewer #4: Yes

5. Is the manuscript presented in an intelligible fashion and written in standard English?

Reviewer #2: Yes

Reviewer #3: Yes

Reviewer #4: Yes

6. Review Comments to the Author

Reviewer #2: Thank to you for the opportunity to review the manuscript ‘Protective and vulnerability factors for PTSD after preterm delivery’ by Grand et al.. The manuscript describes a correlational study between postpartum PTSD symptoms with neuroticism and extraversion among mothers who gave birth prematurely. While the issue of vulnerability to postpartum PTSD and other psychopathologies is of utmost importance, a correlational study at a single timepoint, with a very small sample, does not qualify as addressing “protective and vulnerability factors”. Unfortunately, my assessment of the manuscripts is that it should be rejected, for the following reasons.

1. As noted above, the design of the study and size of the sample do not qualify as addressing protective and vulnerability factors, which are predictive terms. Also using the phrase “propensity to develop PTSD”, is not accurate – the PPQ assesses the number of current symptoms.

2. The manuscript in general is not well written.

a. The introduction has repetitions (for example the first and second paragraphs).

b. It delves into topics completely irrelevant to the study such as mother-infant bonding, attachment or long-term effects of stress.

c. There several missing citations (e.g., lines 67, 68, 85, 89…).

d. There’s confusion regarding the structure of the methods and results: For example, demographics of the sample and reliabilities of the scales should be in the methods section; providing information on the study and obtaining informed consent is in the procedure, it is not pertinent information that participants received a copy of the informed consent.

e. The sources of the original questionnaires need to be cited, as well as their translations.

f. There’s also inconsistency in the use of the decimal point, in some instances it’s a comma in others it’s a period.

g. The use of the term “control group” is incompatible with a correlational study.

h. Typically, statistics are reported once – either in tables or figures or in text. For example, reporting the beta values of the regression, can be either in the table or in the text. The table can contain other information (e.g., B, SE, t, VIF etc.).

3. Results section:

a. In the first part the sample is divided by PPQ cutoff, but this cutoff is not used for hypothesis testing, so why is it necessary? Also, there’s no citation for the cutoff of 6.

b. There are statements in the results section that should be removed completely (e.g., 23% is nearly a quarter), or addressed only in the discussion section (e.g., lines 239-240).

c. Testing the difference in PPQ between women with or w/o a history of depression is not a hypothesis, why was this done?

d. In contrast, after testing whether demographic measures correlated with PPQ, those that were significant could and should have been included in the regression.

Given the small size of the study, the simple replication of previous studies and poor methodological execution I think the manuscript should be rejected.

Reviewer #3: The authors have adequately addressed the comments raised in the previous round of review as seen in the manuscript and author's responses attached to the submitted manuscript. The manuscript describes a technically sound piece of scientific research and conclusions have been drawn from the data presented. The key results as indicated are the protective and vulnerability factors for PTSD among mothers with preterm babies. The study has highlighted these factors well and in the discussion they (factors) have been given as the possible reasons for and against PTSD. The results presented are of immediate interest to many people in my own discipline, or to people from several disciplines since the study highlights protective and vulnerability factors associated with PTSD among mothers with preterm babies, a disorder that has been neglected for a long time yet it is detrimental

All the data has been included in supporting information under database and is available without restriction

Reviewer #4: Thank you for this paper on Protective and vulnerability factors for PTSD after preterm delivery. This is an important topic, and as you pointed out, has been understudied.

TITLE

1. Please include that your study assessed personality traits, perhaps “Protective and vulnerability personality traits associated with PTSD diagnosis after preterm delivery.”

INTRODUCTION

1. Line 69 – I’m not sure that cerebral immaturity is a reason for stress in parents of preterm infants, all babies are cerebrally immature. Do you perhaps mean the potential for neurodevelopmental adverse outcomes in the preterm period? Please explain this thought more.

2. Line 84-86 – stress has also been shown to cause premature activation of the hypothalamic pituitary axis, and may be a risk factor for preterm birth, I think this should be addressed/discussed as well.

3. Line 97-98, you note that ”Research on this subject estimates that it occurs in 25% to 33% of mothers after a preterm birth.” What were the risk factors for the development of PTSD in mothers who experienced preterm birth. Please add this to the discussion here, as it makes the reader interested to find out more from your study.

4. Line 132, you note that ”Certain personality traits, notably extraversion and 133 neuroticism, are strongly correlated with the emergence of anxiety disorders.” However, there is no citation for this statement. Please include one as well as additional rationale for choosing only these 2 of the 5 personality traits to include in this study.

MATERIALS AND METHODS

1. Line 154 – you note that mothers whose babies died were not included in the study. Please discuss your rationale for this decision. Also I think you should discuss later how this may have affected your results. By excluding participants whose babies died, I think you are skewing the results. People whose babies died are more likely to experience PTSD from the event.

2. Please include the validity, sensitivity/specificity and positive/negative predictive value of the PPQ and NEO-FFI surveys.

3. Was a sample size calculated for this study? Please include this information.

DISCUSSION

1. Please include a discussion of how the results may have been impacted by the exclusion of mothers whose babies died.

2. I think you should also discuss how your results were likely influenced by the inclusion of mothers of preterm infants born at < 33 weeks (morse severe than infants born between 34 and 36 weeks (late preterm)

7. PLOS authors have the option to publish the peer review history of their article (what does this mean?). If published, this will include your full peer review and any attached files.

Reviewer #2: No

Reviewer #3: **Yes: **BEATRICE MUKABANA

Reviewer #4: No

---

## [Author Response · Author response to Decision Letter 1]

8 Jan 2024

Dear Reviewer,

Thank you for your recommendation on our paper. We really appreciate the time and consideration you put into our work. Here are our answers to the questions and remarks you raised. Each reviewer will be addressed individually.

We considered the comments of reviewer 2 and edited our paper as per his or her suggestion when it was possible. We modified the title, added some citations, made sure we were consistent in writing our decimals and rewrote some part in order to improve our paper. Although we are sorry our study does not meet your expectations, we appreciate your recommendation to improve it.

Reviewer 3:

Thank you very much for your comment and the time you spent reviewing our paper. We are glad our paper met your expectations and agree on the interest and importance this field of research has. 

Reviewer 4:

Thank you for your time and the comments you made in order to improve our paper. Here are our responses to each of your point:

TITLE

1. Please include that your study assessed personality traits, perhaps “Protective and vulnerability personality traits associated with PTSD diagnosis after preterm delivery.”

We modified the title as suggested to better reflect our study.

INTRODUCTION

1. Line 69 – I’m not sure that cerebral immaturity is a reason for stress in parents of preterm infants, all babies are cerebrally immature. Do you perhaps mean the potential for neurodevelopmental adverse outcomes in the preterm period? Please explain this thought more.

Maybe we were not clear enough. Although we do not think cerebral immaturity is the cause of stress in parents of preterm infants, research show that those babies are less likely to respond back to social cues which is highly important in the development of the bond. 

2. Line 84-86 – stress has also been shown to cause premature activation of the hypothalamic pituitary axis, and may be a risk factor for preterm birth, I think this should be addressed/discussed.

This is a very interesting point that we have considered. Although we are aware of studies such as the study of Julia Seng (among others) which indeed found a correlation between prenatal stress in mothers and preterm birth, we decided not to add this perspective in our paper. Mothers who gave birth prematurely already show intense feelings of guilt and as clinicians, we did not want to add to it. Moreover, although it would have been very interesting to measure the stress in mothers before they gave birth (at T0) to have a comparison level, it would have been logistically very complicated. It would imply meeting every pregnant woman not knowing if they are actually going to give birth prematurely and be included in the study. 

We have added this limit to our study in the discussion part. Thank you for this feedback.

3. Line 97-98, you note that ”Research on this subject estimates that it occurs in 25% to 33% of mothers after a preterm birth.” What were the risk factors for the development of PTSD in mothers who experienced preterm birth. Please add this to the discussion here, as it makes the reader interested to find out more from your study.

We did! You are right. Thank you for your comment.

4. Line 132, you note that ”Certain personality traits, notably extraversion and 133 neuroticism, are strongly correlated with the emergence of anxiety disorders.” However, there is no citation for this statement. Please include one as well as additional rationale for choosing only these 2 of the 5 personality traits to include in this study.

We tested all 5 personality traits in our study but only the two we had expected (extraversion and neuroticism) showed a correlation. Our hypotheses only focused on these two as they are often found in other studies as correlated with our interest here (depression, anxiety, etc). Citations 24 to 26 are studies that show correlations between those two traits and the development of pathologies like depression or anxiety and level of stress. I added a line in the discussion to clarify that we did assess all five dimensions in our study.

MATERIALS AND METHODS

1. Line 154 – you note that mothers whose babies died were not included in the study. Please discuss your rationale for this decision. Also I think you should discuss later how this may have affected your results. By excluding participants whose babies died, I think you are skewing the results. People whose babies died are more likely to experience PTSD from the event.

This is indeed a decision we made for several reasons:

- Women whose infants died do not stay at the hospital and we would have been very difficult for us to include them for the study

- For ethical reasons and because we are clinicians, it seems to us that taking part in a research shortly after experience child loss would not have been beneficial for women

- Finally, comparing women whose baby died with women whose baby did not could be another study by itself. One is experiencing symbolic mourning (of the ideal baby) while the other is experiencing real grief (real as in “the reality”).

I added it in the discussion section to help the reader understand our choice.

2. Please include the validity, sensitivity/specificity and positive/negative predictive value of the PPQ and NEO-FFI surveys.

The internal consistency is high. Our Cronbach’s alphas are consistent with those of the original test (ranging from .68 for Agreeableness to .89 for Neuroticism). A two-week retest reliability was made for the NEO-FFI in 2001 (Robins & al., 2001) and showed scores from .86 to .90. The NEO inventories scales have been widely used to measure personality. 

As for the PPQ, the internal consistency is good with Cronbach’s alphas of .73 and .74 (internal consistency of the two factors in the PPQ). The retest reliability done one month apart is .92. Its validity has been measured by comparing the PPQ with IES (Impact of Event Scale) obtaining a .78 correlation between the two instruments. (Pierrehumbert & al., 2004).

3. Was a sample size calculated for this study? Please include this information.

There was no sample size calculated for this study as we conducted a primary and prospective survey in 2017 which already showed promising results and the same correlations. Hence our wish to continue the study and to include as many participants as we could that following year. 

DISCUSSION

1. Please include a discussion of how the results may have been impacted by the exclusion of mothers whose babies died.

We added the rationale behind it in the discussion as suggested.

3. I think you should also discuss how your results were likely influenced by the inclusion of mothers of preterm infants born at < 33 weeks (morse severe than infants born between 34 and 36 weeks (late preterm)

You are right. We made the choice of focusing on mothers of preterm infants born at < 33 weeks because they stay at the hospital longer and can be more easily included in research. I will specify it throughout the paper as saying “preterm birth” is not precise enough as we only focused on severe prematurity. Thank you.

---

## [Decision Letter · Decision Letter 2]

14 May 2024

PONE-D-22-23913R2Protective and vulnerability personality traits associated with PTSD diagnosis after preterm deliveryPLOS ONE

Dear Dr. Eutrope,

Thank you for submitting your manuscript to PLOS ONE. After careful consideration, we feel that it has merit but does not fully meet PLOS ONE’s publication criteria as it currently stands. Therefore, we invite you to submit a revised version of the manuscript that addresses the points raised during the review process.

We look forward to receiving your revised manuscript.

Kind regards,

Giuseppe Marano

Academic Editor

PLOS ONE

Reviewers' comments:

Reviewer's Responses to Questions

**Comments to the Author**

1. If the authors have adequately addressed your comments raised in a previous round of review and you feel that this manuscript is now acceptable for publication, you may indicate that here to bypass the “Comments to the Author” section, enter your conflict of interest statement in the “Confidential to Editor” section, and submit your "Accept" recommendation.

Reviewer #5: All comments have been addressed

Reviewer #6: (No Response)

Reviewer #7: All comments have been addressed

2. Is the manuscript technically sound, and do the data support the conclusions?

Reviewer #5: Partly

Reviewer #6: Yes

Reviewer #7: Yes

3. Has the statistical analysis been performed appropriately and rigorously? 

Reviewer #5: No

Reviewer #6: Yes

Reviewer #7: Yes

4. Have the authors made all data underlying the findings in their manuscript fully available?

Reviewer #5: Yes

Reviewer #6: Yes

Reviewer #7: Yes

5. Is the manuscript presented in an intelligible fashion and written in standard English?

Reviewer #5: Yes

Reviewer #6: Yes

Reviewer #7: Yes

6. Review Comments to the Author

Reviewer #5: 1. In Table 4, Cohens d value has not been included.

2. in Table 4 confidence intervals of upper and lower limit also not included.

3. In table 5 confidence intervals of upper and lower limit also not included.

4. In table 5, F value not mentioned.

5. In Table 5 Both Unstandardized and standardized beta coefficients have to be reported

Reviewer #6: I appreciate the opportunity to review this manuscript. However, there are a few issues that need to be addressed.

please delete the question "What were the risk factors for the development of PTSD..." and discuss the risk factors associated with PTSD Development as recommended by previous reviewer 4.

Before line 128, it would be appropriate for the authors to provide two or three statements on the findings of previous empirical studies on personality traits in this context. This might justify why they choose neuroticism and extraversion as recommended by previous Reviewer 4.

It is kindly recommended that the authors also discuss the results of the second hypothesis (comparing PTSD Mothers and Non-PTSD Mothers concerning history of depression) in the discussion section.

Thank you.

Reviewer #7: This study aimed to investigate the relationship between personality traits and post-traumatic stress disorder (PTSD) in mothers who experienced preterm delivery. The research question and objectives were clear and well-defined, and the literature review provided a solid foundation for the study. The methodology and statistical analysis were well-explained and appropriate for the research question.

The study's findings on the correlation between neuroticism and extraversion with PTSD were interesting and relevant. However, the study had several limitations that should be addressed in future research. The small sample size (51 participants) and specific population studied (mothers who gave birth prematurely before 33 weeks) limit the generalizability of the findings. The lack of a control group or comparison with full-term mothers makes it difficult to determine if the findings are unique to preterm delivery. Additionally, only two personality traits were found to be correlated with PTSD, while other traits were not explored.

The study's reliance on self-reported measures may be subject to bias, and the lack of longitudinal follow-up to assess the long-term effects of PTSD on mothers and children is a significant limitation. Furthermore, other potential explanatory variables, such as social support, coping mechanisms, or obstetric history, were not considered.

Recommendations:

- Increase the sample size and diversify the population studied to enhance generalizability.

- Include a control group or comparison with full-term mothers to determine if the findings are unique to preterm delivery.

- Explore additional personality traits and potential explanatory variables.

- Use multiple measures, including clinical interviews, to reduce bias.

- Conduct longitudinal follow-up to assess the long-term effects of PTSD on mothers and children.

Strengths:

1. Clear research question and objectives

2. Relevant literature review on PTSD, preterm delivery, and personality traits

3. Well-explained methodology and statistical analysis

4. Interesting and relevant findings on the correlation between neuroticism and extraversion with PTSD

5. Discussion of limitations and potential avenues for future research

Weaknesses:

1. Small sample size (51 participants)

2. Limited generalizability due to the specific population studied (mothers who gave birth prematurely before 33 weeks)

3. No control group or comparison with full-term mothers

4. Only two personality traits (neuroticism and extraversion) were found to be correlated with PTSD, while other traits (openness, conscientiousness, and agreeableness) were not

Add these limitations as future recommendations

1. Longitudinal follow-up to assess the long-term effects of PTSD on mothers and children

2. Consideration of other potential explanatory variables, such as social support, coping mechanisms, or obstetric history

3. The study relied on self-reported measures, which may be subject to bias

Overall, the article presents a clear and well-structured research study, but with some limitations that should be addressed in future research. The findings on the correlation between personality traits and PTSD are interesting and relevant, but require further exploration and replication with larger and more diverse samples.

7. PLOS authors have the option to publish the peer review history of their article (what does this mean?). If published, this will include your full peer review and any attached files.

Reviewer #5: No

Reviewer #6: No

Reviewer #7: **Yes: **Dr. Syeda Rubab Aftab

---

## [Author Response · Author response to Decision Letter 2]

29 Jun 2024

Dear Reviewers,

Thank you for the time spent on reviewing our paper. We edited our manuscript based on your suggestions. 

Reviewer 5:

- Table 4. We added the Cohen d value and confidence intervals as suggested. The Cohen d value for this analysis was .83 and confidence intervals of 95% limits were [0,32 ; 3,92]. Everything was added in the paper.

- Table 5. We also added the F of the multiple regression which is 2,64 (p=.04) as well as the unstandardized betas.

Reviewer 6:

- I deleted the question as suggested and added a few lines to introduce the next paragraph.

- There are no prior study studying the link between personality traits and PTSD. As explained in our paper, past research studied external factors (baby weight, etc) and found no correlation. We then decided to focus on internal factors such as personality traits. We later explained how personality traits can be correlated to such pathologies mentioning several studies on that matter. I am not sure what or how to change.

- I added a paragraph to further develop our ideas on the topic. Thank you! 

Reviewer 7:

- Thank you for your feedback, however if I may correct one thing: all five traits were explored but only two were significant. We mentioned that we did test all five of them (line 226).

- The environmental factors that you mentioned are very interesting and we did test some of them (social support or obstetric history for example but they were not significant, line 231). 

- I understand your idea for a follow-up study on the effects of PSTD which I find very interesting. However, there are already some research of the topic and we decided to focus on the vulnerability and protections factors to PTSD, not PTSD itself for our research. Our main goal was to better identify women at risk and prevent the risk for PTSD.

- Your suggestions are very appreciated, I added them on our discussion and will keep them in mind for future research!

---

## [Decision Letter · Decision Letter 3]

25 Jul 2024

Protective and vulnerability personality traits associated with PTSD diagnosis after preterm delivery

PONE-D-22-23913R3

Dear Dr. Eutrope,

We’re pleased to inform you that your manuscript has been judged scientifically suitable for publication and will be formally accepted for publication once it meets all outstanding technical requirements.

Kind regards,

Giuseppe Marano

Academic Editor

PLOS ONE

Additional Editor Comments (optional):

Reviewers' comments:

Reviewer's Responses to Questions

**Comments to the Author**

1. If the authors have adequately addressed your comments raised in a previous round of review and you feel that this manuscript is now acceptable for publication, you may indicate that here to bypass the “Comments to the Author” section, enter your conflict of interest statement in the “Confidential to Editor” section, and submit your "Accept" recommendation.

Reviewer #5: All comments have been addressed

Reviewer #6: All comments have been addressed

2. Is the manuscript technically sound, and do the data support the conclusions?

Reviewer #5: Yes

Reviewer #6: Yes

3. Has the statistical analysis been performed appropriately and rigorously? 

Reviewer #5: Yes

Reviewer #6: Yes

4. Have the authors made all data underlying the findings in their manuscript fully available?

Reviewer #5: Yes

Reviewer #6: Yes

5. Is the manuscript presented in an intelligible fashion and written in standard English?

Reviewer #5: Yes

Reviewer #6: Yes

6. Review Comments to the Author

Reviewer #5: The article is now in appropriate form to be published. Check thoroughly once the references and in text citations.

Reviewer #6: I am thankful for the chance to review this manuscript. The authors have addressed the concerns raised in the previous submission. I look forward to seeing it in print.

7. PLOS authors have the option to publish the peer review history of their article (what does this mean?). If published, this will include your full peer review and any attached files.

Reviewer #5: No

Reviewer #6: No

---

## [Editor Report · Acceptance letter]

2 Aug 2024

PONE-D-22-23913R3 

PLOS ONE

Dear Dr. Eutrope, 

I'm pleased to inform you that your manuscript has been deemed suitable for publication in PLOS ONE. Congratulations! Your manuscript is now being handed over to our production team.

Kind regards, 

on behalf of

Dr. Giuseppe Marano 

Academic Editor

PLOS ONE